# Association of Dietary Intakes and Genetically Determined Serum Concentrations of Mono and Poly Unsaturated Fatty Acids on Chronic Kidney Disease: Insights from Dietary Analysis and Mendelian Randomization

**DOI:** 10.3390/nu14061231

**Published:** 2022-03-15

**Authors:** Mohsen Mazidi, Andre P. Kengne, Mario Siervo, Richard Kirwan

**Affiliations:** 1Medical Research Council Population Health Research Unit, University of Oxford, Oxford OX3 7LF, UK; 2Clinical Trial Service Unit and Epidemiological Studies Unit (CTSU), Nuffield Department of Population Health, University of Oxford, Oxford OX3 7LF, UK; 3Department of Twin Research & Genetic Epidemiology, South Wing St Thomas’, King’s College London, London SE1 7EH, UK; 4Non-Communicable Disease Research Unit, South African Medical Research Council, University of Cape Town, Cape Town 7505, South Africa; andre.kengne@mrc.ac.za; 5School of Life Sciences, Queen’s Medical Centre, The University of Nottingham Medical School, Nottingham NG7 2RD, UK; mario.siervo@nottingham.ac.uk; 6School of Biological and Environmental Sciences, Liverpool John Moores University, Liverpool L3 3AF, UK; r.p.kirwan@2018.ljmu.ac.uk

**Keywords:** mendelian randomization, serum fatty acids, monounsaturated fatty acids, chronic kidney disease, polyunsaturated fatty acids, renal function

## Abstract

Polyunsaturated fatty acid (PUFA) intake is generally associated with better renal function, while the association of monounsaturated fatty acids (MUFAs) remains unconfirmed. Mendelian randomization (MR) analysis was used to obtain unconfounded estimates of the causal association of dietary intake and genetically determined serum PUFA and MUFA levels with measures of renal function. Data from participants of the National Health and Nutrition Examination Surveys (NHANES) from 2005 to 2010 were used. Data from the largest genome-wide association studies (GWAS) on MUFAs, PUFAs, eGFR, and chronic kidney disease (CKD) were analysed for the entire sample. A total of 16,025 participants were included. eGFR improved across increasing quartiles of total PUFA intake from 86.3 ± 0.5 (Q1) to 96.2 ± 0.5 mL/min/1.73 m² (Q4), (*p* < 0.001). Conversely, there was no association between MUFA intake and measures of renal function (all *p* > 0.21). In multivariable models, the top quartile of PUFA intake had a 21% lower risk for CKD, but there was no significant association between CKD risk and MUFA intake. Genetically determined serum MUFA (heptadecenoate (17:1), myristoleic acid (14:1), and palmitoleic acid (16:1)) and PUFA (α-linolenic acid and eicosapentaenoic acid) concentrations had no significant association with eGFR and CKD risk. Additionally, no association was found in the analyses stratified by diabetes status. Higher dietary PUFA intake is associated with lower risk of CKD, while there was no association with serum levels of MUFAs or PUFAs. Additional studies including clinical trials are warranted.

## 1. Introduction

Chronic kidney disease (CKD) is defined as significantly impaired kidney function, identified by a reduced glomerular filtration rate (GFR) or increased urinary albumin excretion (albuminuria) that are confirmed on two or more occasions at least 3 months apart [1]. The global prevalence of CKD has been estimated between 11 to 13%, with prevalence increasing with age, and up to 34% of subjects older than 70 years old have a low eGFR indicating the presence of CKD [2,3]. CKD is known to be associated with and contribute to the disease burden of multiple comorbidities including diabetes [4,5,6], hypertension [7,8], obesity [9,10], and especially cardiovascular diseases [11,12,13,14], and as such, it is associated with a higher all-cause mortality [13].

Both low plasma concentrations and dietary intakes of n-3 and n-6 polyunsaturated fatty acids (PUFA) have previously been associated with impaired renal function [15,16,17], while a lower saturated fatty acid (SFA) intake has been associated with improved renal function [18]. The association between monounsaturated fatty acids (MUFA) and risk of CKD, however, remains poorly understood. A potential mechanism through which PUFAs may play a protective role in kidney function is by downregulating certain aspects of the inflammatory response, for example, a reduction in proinflammatory cytokines [19]. Indeed, lower levels of plasma markers of chronic inflammation such as CRP and tumor necrosis factor alpha have been observed in older adults with higher serum levels of PUFAs [20]. However, such observational data cannot be used to determine the causality of serum PUFAs in the etiology of CKD.

Mendelian randomization (MR) analysis using functional single nucleotide polymorphisms (SNPs) associated with specific changes in physiological exposures (such as serum MUFAs and PUFAs) as genetic instruments of analysis are capable of providing unbiased and robust evidence on the mechanisms of the pathogenesis of disease and the efficacy of treatments. Compared with conventional risk-factor epidemiology, these studies are considerably less prone to confounding, residual bias, and reverse causation [21]. While randomized controlled trials (RCTs) are considered useful for the determination of causality, they are often limited by cost, time, and ethical constraints, depending on the characteristics of the exposure and disease state being studied. MR studies can be considered a way to avoid these inherent issues with RCTs and, in addition to this, the data from such studies can be used to improve the development of pilot RCTs and strategies for clinical trials by elucidating the potential efficacy of an intervention or even the magnitude of effect in selected individuals and groups [22].

Therefore, national survey data (Nutrition and Health Examination Surveys (NHANES)) and Mendelian randomization (MR) analysis were used to determine unbiased estimates of the casual association of genetically determined serum levels and dietary intake of MUFAs and PUFAs with renal function.

## 2. Materials and Methods

### 2.1. National Survey

#### 2.1.1. Population

We used data from the NHANES, which were previously published in detail [23]. In brief, repeated cross-sectional surveys are conducted by the US National Center for Health Statistics (NCHS). These consist of home visits where questionnaires are used to collect data on demographics and health habits such as diet. Complex multistage probability sampling procedures are employed by NHANES to ensure adequate racial/ethnic representation, as well as recruitment from diverse locations [23]. The NCHS Research Ethics Review Board approved the protocol and all participants provided informed consent.

The methods for the specific analyses can be found in the NHANES Laboratory/Medical Technologists Procedures Manual [24,25,26,27]. Blood was drawn from an antecubital vein following a standard protocol. The Jaffe rate method (kinetic alkaline picrate) was used in the D×C800 modular chemistry side in order to determine the concentration of creatinine in the serum. The creatinine calibration was traceable to an isotope dilution mass spectrometry reference method [28]. Urinary creatinine and urinary albumin (which was assessed using a solid-phase fluorescent immunoassay of a random urine sample) [29] were used to calculate the urine albumin to creatinine ratio (ACR). eGFR was calculated using the CKD Epidemiology Collaboration (CKD-EPI) equation (in mL/min/1.73 m²). Prevalent CKD was identified as an eGFR <60 mL/min/1.73 m² [29].

#### 2.1.2. Dietary Intake

Dietary intake was assessed though 24 h recall, obtained with the assistance of a trained interviewer, through the use of the United States Department of Agriculture Automated Multiple-Pass Method (AMPM), which is an interview system with standardized probes that is aided by a computerized system [30,31]. Briefly, the type and quantity of all foods and drinks ingested in the 24 h period prior to the dietary interview (from 12 am to 12 am) were collected using the AMPM. The AMPM is designed to ensure more complete and accurate data collection while reducing the burden on participants [31,32]. In the current study, we used the data on the total MUFA intake as well as the intake of specific MUFAs: 16:1 (Hexadecenoic, commonly known as palmitoleic acid), 18:1 (Octadecenoic, commonly known as oleic acid (OA)), 20:1 (Eicosenoic), and 22:1 (Tetracosenoic). We also analyzed data on the intake of total PUFAs, as well as the individual PUFAs: PUFA 18:2 (octadecadienoic, commonly known as linoleic acid (LA)), PUFA 18:3 (octadecatrienoic, commonly known as α-linolenic acid (ALA)), PUFA 18:4 (octadecatetraenoic), PUFA 20:4 (eicosatetraenoic (ETA)), PUFA 20:5 (eicosapentaenoic (EPA)), PUFA 22:5 (docosapentaenoic (DPA)), and PUFA 22:6 (docosahexaenoic (DHA)).

#### 2.1.3. Statistical Analysis

CDC guidelines for the analysis of complex NHANES data, using the appropriate weighting methods and accounting for the masked variance, were employed [33]. Mean and standard error of mean (SEM) were used for continuous (analysis of variance) measures and percentage for categorical variables (chi-square). In order to evaluate the normality of data, the Kolmogorov−Smirnov test was applied. The adjusted mean of specific kidney function markers (serum creatinine, ACR, and eGFR) across MUFA and PUFA quartiles were estimated using analysis of covariance (ANCOVA). These models were adjusted for age, sex, race, poverty to income ratio, fasting blood glucose, systolic and diastolic blood pressure, energy intake, red meat intake, body mass index (BMI, kg/m^2^), diabetes (DM) (self-reported history of DM or fasting plasma glucose ≥126 mg/dL), and hypertension (HTN), diagnosed in individuals with systolic blood pressure ≥140 mmHg, diastolic blood pressure ≥90 mmHg, or in those on antihypertensive drugs [34]. Log transformations were performed for variables with departure from normal distribution. Logistic regressions models with three different levels of adjustments (model 1: age, sex, race and poverty to income ratio; model 2: age, sex, race, poverty to income ratio, fasting blood glucose, systolic and diastolic blood pressure, and hypertension (HTN); and model 3: age, sex, race, poverty to income ratio, fasting blood glucose, systolic and diastolic blood pressure, HTN, triglycerides (TG) and high density lipoprotein cholesterol (HDL), diabetes mellitus (DM), body mass index (BMI), and C-reactive protein (CRP)), were then used to derive the odds ratio (OR) and 95% confidence interval (CI) for the association with prevalent CKD across MUFA and PUFA quartiles. The lowest quartile was always used as the reference value. Variance inflation factors (VIF) were applied at each step to assess multi-collinearity for the multiple linear regressions [35]. Multi-collinearity was considered high when the VIF was greater than 10 [35]. All of the tests were two sided, and a *p*-value of less than 0.05 characterized significant results.

### 2.2. Mendelian Randomization

#### 2.2.1. Study Design

This study employed a two-sample MR study design. Briefly, summary statistics from different genome wide association studies (GWAS) were analyzed for the exposures (serum MUFAs and PUFAs) and outcomes (renal function) of interest, to estimate the effects of the former on the latter [36]. That is to say, genetic predictors of serum MUFAs and PUFAs were applied to extensively genotyped case-control studies of renal function (eGFR and the risk of CKD) to obtain estimates of their association.

#### 2.2.2. Genetic Predictors of Exposures

We retrieved summary data for the association between SNPs and four circulating MUFAs (myristoleic acid (14:1 (tetradecenoic)), palmitoleic acid (16:1 (hexadecenoic)), 10-heptadecenoate (17:1), oleic acid (18:1 (octadecenoic))) from the GWAS (7824 adults of European ancestry) (Appendix A). Genotyping, quality control, and imputation procedures have been previously elaborated [37]. For analysis of the serum PUFAs, we retrieved summary data for the association between SNPs and circulating ALA and EPA concentration from the CHARGE meta-GWAS (*n* = 8866 adults, European descent) [38]. In instances where an SNP was not available for the outcome GWAS summary statistics, proxy SNPs were identified. A minimum linkage disequilibrium (LD) r^2^ = 0.8 was required for such proxy SNPs. Bias in effect estimates can induced by correlation between SNPs, and in order to minimize this bias, our genetic instruments were limited to independent SNPs not in linkage disequilibrium (*p* = 0.0001). We refer to a set of SNPs that proxy serum MUFAs and PUFAs as “genetic instruments”.

#### 2.2.3. Genetic Predictors of Outcomes

A meta-analysis consisting of the largest genotyped study sample (*n* = 133,413 with replication in up to 42,166 participants) was used to obtain genetic associations with eGFR [39], which was determined using the four-variable Modification of Diet in Renal Disease (MDRD) Study Equation [39]. The determination of CKD was an eGFR <60 mL/min/1.73 m^2^ and that of type 2 diabetes (T2 D) was fasting glucose ≥126 mg/dL, antidiabetic drug treatment, or self-reported. Kidney function and T2 D were assessed simultaneously. For the GWAS analysis, a centralized analysis plan was applied with each study regressing sex- and age-adjusted residuals of the logarithm of eGFR on the SNP dosage levels. Furthermore, logistic regression of CKD was performed on SNP dosage levels, adjusting for sex and age. For all traits, adjustment for appropriate study-specific features, such as study site and genetic principal components, was included in the regression, and family-based studies appropriately accounted for relatedness.

#### 2.2.4. Statistics

We combined the effect of instruments using the inverse variance weighted (IVW) method. The Q value for IVW was used to determine heterogeneity. As the final effect estimate may be potentially affected by pleiotropic variants, a sensitivity analysis including weighted median (WM) and MR-Egger and using the leave-one-out method, was performed [36]. Causal estimates in MR-Egger are less precise than those obtained using IVW MR [40] due to a lower false-positive rate and an associated higher false-negative rate, leading to a lower statistical power [41].

The Q′ heterogeneity statistic was used to determine the heterogeneity between individual genetic variant estimates [42]. Furthermore, the instrumental variable analysis “exclusion-restriction” assumption was assessed by using Ensembl release (http://useast.ensembl.org/index.html, accessed on 2 April 2020) and PhenoScanner (SNP phenotypes are provided by Ensembl and the phenotypes of correlated SNPs are provided by PhenoScanner).

#### 2.2.5. Sensitivity Analysis

Sensitivity analysis was performed using MR-Egger and the MR pleiotropy residual sum and outlier (MR-PRESSO) test [42]. Outlier effect estimates were detected by the MR-PRESSO framework, which subsequently removed them from the analysis. This was done by regression of the variant−outcome associations on variant−exposure associations. Furthermore, the MR-Robust Adjusted Profile Score (RAPS) was applied in order to correct for pleiotropy. To qualify as a result, the causal estimates must agree in both direction and magnitude across MR methods, must have nominal significance in IVW MR, and must not show any evidence of bias from horizontal pleiotropy. All analyses were done using R software (version 3.4.2 R Core Team, 2017).

## 3. Results

### 3.1. Dietary Intake

A total of 16,025 of the NHANES participants fulfilled the criteria for inclusion in the analysis; 6.8% had prevalent CKD. The characteristics of participants for the whole sample and by CKD status are summarized in Table 1. Overall, 51.2% of the participants were women, with no significant gender difference observed between those presenting with or without CKD (*p* = 0.412). Compared with those without CKD, participants with CKD were more likely to be non-Hispanic Whites (82.9 vs. 68.4%), and less likely Mexican-Americans (2.5 vs. 9.0%), non-Hispanic Black (8.1 vs. 11.2%), other Hispanic (3.0 vs. 5.4%); *p* < 0.001 for differences in the distribution of ethnicity by CKD status. The mean age was 45.8 years; participants with CKD were older than those without (69.1 vs. 44.6 years, *p* < 0.001). Those presenting with CKD had a higher BMI (*p* < 0.001) and higher serum CRP, TG, and fasting glucose (*p* < 0.001 for all comparisons), as well as being more likely to have DM and HTN (*p* < 0.001 for all comparisons, Table 1).

Adjusted (age, sex, race, fasting blood glucose, systolic and diastolic blood pressure, energy intake, red meat intake, BMI, DM, and HTN) mean levels of kidney function by quartiles of total MUFAs and PUFAs are shown in Table 2. Across increasing the total PUFA quartiles, renal function improved, with, for example, the adjusted mean of eGFR changing from 86.3 ± 0.5 in Q1 to 96.2 ± 0.5 mL/min in Q4 (*p* < 0.001). Across increasing the total MUFA quartiles, the mean urine albumin, log ACR, and eGFR did not change significantly (all *p* > 0.213, Table 2). Across quartiles of different types of MUFAs (hexadecenoic (16:1), octadecenoic (18:1), eicosenoic (20:1), and docosenoic (22:1)) and PUFAs (PUFA 18:2 (octadecadienoic), PUFA 18:3 (octadecatrienoic), PUFA 18:4 (octadecatetraenoic), PUFA 20:4 (eicosatetraenoic), PUFA 20:5 (eicosapentaenoic), PUFA 22:5 (docosapentaenoic), and PUFA 22:6 (docosahexaenoic)) urine albumin, log ACR, and eGFR did not change significantly (all *p* > 0.412, data not shown). Multiple potential confounders, arranged into three separate models, were used to evaluate the odds of CKD across the quartiles of the total MUFAs and PUFAs (Table 3). In the model adjusted for age, sex, race, and poverty to income ratio, compared with the lowest quartile of the total PUFA, the OR (95%CI) for CKD was 0.60 (0.40–0.81) for the top quartile (*p* = 0.012 for trend, Table 3).

Employing the expanded models and following adjustment for age, sex, race, poverty to income ratio, fasting blood glucose, systolic and diastolic blood pressure, HTN, DM, TG, HDL, and CRP the top quartile of total PUFA intake had a 21% lower likelihood of CKD (OR = 0.79 (0.68–0.88)). With regard to MUFA intake, in three different models with range-varied confounders (age, sex, race, poverty to income ratio, fasting blood glucose, systolic and diastolic blood pressure, HTN, DM, TG, HDL, and CRP), we found no association between intake of MUFA and prevalent CKD (Table 3).

### 3.2. Mendelian Randomization

The instrument associations for serum MUFA and PUFA levels are shown in Appendix A. The instruments had F-statistics ranging from 142 to 236, making significant bias from use of weak instruments unlikely (42). The results, expressed as beta-coefficient for serum MUFA/PUFA per 1 standard deviation (SD) increase in outcomes, are presented in Appendix A.

#### 3.2.1. MUFAs

Genetically higher serum heptadecenoate (17:1), myristoleic acid (14:1), oleic acid (18:1), and palmitoleic acid (16:1) levels had no significant effect on risk of CKD (IVW = Beta: >0.2052, *p* > 0.5888, Appendix A) or level of eGFR (IVW = Beta: >−0.0098, *p* > 0.6569, Appendix A). Genetically determined levels of serum heptadecenoate (17:1), myristoleic acid (14:1), oleic acid (18:1), and palmitoleic acid (16:1) had no significant impact on CKD in either diabetic subjects (IVW = Beta: >−0.00512, *p* > 0.9542, Appendix A) or non-diabetic subjects (IVW = Beta: >−0.00174, *p* > 0.937, Appendix A).

#### 3.2.2. PUFAs

With regards to the impact of PUFAs on renal function, we found that genetically higher levels of serum ALA and EPA had no significant impact on the risk of CKD (IVW = Beta: 0.3791, *p* > 0.8125), nor the level of eGFR (IVW = Beta: −0.04827, *p* > 0.486). Genetically determined levels of serum ALA and EPA had no significant impact on either diabetic subjects (IVW = Beta: −0.3987, *p* > 0.1593, Appendix A) or normal subjects (IVW = Beta: −0.03081, *p* > 0.6565, Appendix A).

Heterogeneity analyses and pleiotropy bias are also shown in Appendix A. The estimation based on both MR Egger and IVW was higher than 0.05, which indicted a low chance of heterogeneity (all IVW *p* > 0.075, all MR Egger *p* > 0.063). Furthermore, the results of the MR-PRESSO did not indicate outliers for all of the estimates. The horizontal pleiotropy test, with very negligible Egger regression intercept, also indicated a low likelihood of pleiotropy for all of our estimations (all *p* > 0.212). The results of the MR-RAPS were identical with the IVW estimates, again highlighting a low likelihood of pleiotropy (Appendix A). The results of the leave-one-out method demonstrated that the links were not driven by single SNPs. A graphical summary of methodology and results is displayed in Figure 1.

## 4. Discussion

In this article, we analyzed dietary data on MUFAs and PUFAs intake, along with a set of genetic variants that have been demonstrated to be associated with four circulating serum MUFAs and PUFAs in order to determine their association with renal function. No significant association was observed between different dietary intakes of MUFAs. Conversely, the dietary intake of PUFAs was inversely associated with measures of kidney function and prevalent CKD. However, MR analyses did not support any causal effect of various MUFA or PUFA concentrations on CKD.

CKD is commonly observed as a comorbidity in chronic lifestyle diseases, including diabetes, hypertension, obesity, and cardiovascular disease [4,5,6,7,8,9,10,11,12,13,14], all of which are components of metabolic syndrome (MetS), which itself has been associated with CKD in multiple meta-analyses [43,44]. Lifestyle change involving dietary intervention is seen as a potentially cost-effective treatment for MetS [45,46]. In particular, modulation of dietary fatty acids, comprising replacement of saturated fatty acids (SFA) with MUFAs and PUFAs, has been shown to be beneficial for the prevention and improvement of MetS components [47,48,49,50,51]. Due to the prevalence and clinical significance of CKD [1,2], determining the role of specific macronutrients in its etiology should be considered important.

Although the cause−effect association between CKD and certain components of the MetS (diabetes and hypertension) is believed to be bidirectional [11] and higher ratios of MUFA:SFA intake, along with increased intake of PUFAs, can lead to improvement of these conditions [50,51,52], there is little evidence to show a direct association between MUFA intake specifically and CKD. For example, while PUFA intake was associated with a lower risk of CKD in a non-diabetic population, such an association was not observed for MUFA [17]. Similarly, in an observational study in a diabetic population, MUFA intake was not associated with improvements in renal function, while an association was observed for the MUFA:SFA ratio and intakes of PUFAs [18]. There are a number of putative mechanisms through which an increased intake of PUFAs could potentially protect renal function, ranging from effects on regulatory molecules involved in renal inflammatory processes [53], reduction in proteinuria [54], improvement of blood pressure levels [55], reduction of serum triglycerides [56], and even improvements in blood vessel function [57].

We found no significant association between different dietary intakes of total and individual MUFAs and CKD. The addition of MR analysis makes our study superior to simple observational studies, as MR is a powerful tool for the detection of causation [21], and our results do not support a causal association between serum MUFAs and CKD. The MR analysis also showed no such association between genetically determined markers of serum ALA and EPA and renal function. This could potentially indicate that the results of the observational study of dietary intake could be affected by confounding or even reverse causation. Briefly, healthy dietary choices (such as increased PUFA intake) often occur together with other healthy dietary or lifestyle factors [58], leading to confounding of the interpretation of results. For example, healthy dietary patterns such as Mediterranean, DASH, and Prudent, which tend to be rich in fruits, vegetables, and whole grains, and lower in red and processed meat, refined grains, and added sugar, are associated with a reduced risk of chronic diseases such as obesity, diabetes, and cardiovascular disease [59], and subsequently with a reduced risk of CKD [60]. The overall composition of these healthier dietary patterns, as well as other healthy lifestyle behaviors that often occur together [58,61] and that may promote better kidney function, may confound a dietary analysis focused on specific nutrients, in this case PUFAs. Indeed, this may explain the difference in results for the dietary analysis and mendelian randomization in relation to PUFAs and risk of CKD. Similarly, reverse causality, whereby the knowledge of a disease status or disease-marker influences dietary choices, can be particularly problematic in retrospective studies [62]. However, another possible reason for the discrepancy in the dietary analysis and MR results should be considered. It should not be overlooked that serum fatty acids are known to be determined largely due to dietary intake [63]. Therefore, this study may highlight the fact that MR may not be a suitable analysis method for determining the role of such serum markers that are more dependent on diet as opposed to genetics.

Our study has some limitations. Firstly, the inability of the NHANES database to distinguish between MUFAs of an animal and plant origin, which could potentially contribute to the lack of association in this and other observational studies of CKD. In the “Western diet” pattern, MUFAs are predominantly supplied by foods of an animal origin, while in countries that typically follow a Mediterranean diet for example, extra virgin olive oil is the major source of these fatty acids [64]. Accordingly, dietary interventions high in MUFAs of a plant origin (from olive oil or nuts) have been shown to have benefits on cardiometabolic health, including hypertension and glucose control [46,65], whereas studies not differentiating between sources of MUFAs have found no benefit [16]. There is debate as to whether the observed benefits of MUFAs may be due to the fatty acids themselves or to other bioactive components in the fat sources or the dietary patterns [66,67,68,69]. Thus, elucidating the role of plant or animal MUFA intake in this particular cohort is not possible. A further limitation is that MR analysis is known to have limited statistical power, such that a lack of finding in our analysis might be due to small causal effects that were not detectable in our study. Finally, while we used the largest GWAS that is currently available in the literature for this analysis, the availability of future GWAS with a greater sample size, and thus providing more statistical power, may warrant further analysis of this topic at that time.

## 5. Conclusions

In conclusion, we found no evidence to support an association between the intake of total or individual MUFAs, nor a causal effect of serum MUFAs on CKD. While no causal effect of genetically determined serum PUFAs on prevalent CKD could be determined, a clear inverse association was observed between the dietary intake of PUFAs and CKD. While further investigation is required into the role of PUFAs in the development of CKD, these findings do not contradict the current evidence-base for the benefit of replacing dietary SFAs with MUFAs and, preferentially, PUFAs [70].

## Figures and Tables

**Figure 1 nutrients-14-01231-f001:**
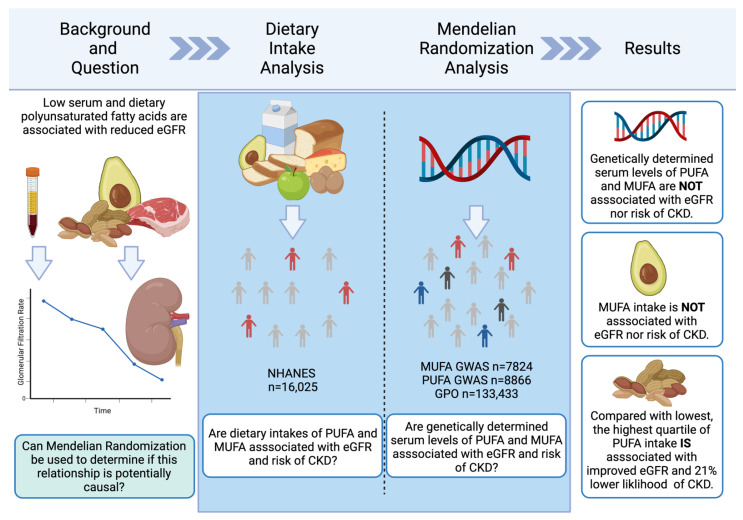
Graphical summary of the study methodology and results. eGFR: estimated glomerular filtration rate; NHANES: Nutrition and Health Examination Survey; PUFA: polyunsaturated fatty acid; MUFA: monounsaturated fatty acid; CKD: chronic kidney disease; GWAS: genome-wide association study; GPO: genetic predictors of outcomes.

**Table 1 nutrients-14-01231-t001:** Demographic characteristics of subjects for the whole sample and stratified by chronic kidney disease (CKD) status.

Characteristics		Overall	With CKD	Without CKD	*p*-Value
Sex	Men (%)	48.8	37.7	49.2	<0.001
	Women (%)	51.2	62.2	50.9	
Age (years), mean (± SEM)		45.8 ± 0.1	69.1 ± 0.2	44.6 ± 0.2	<0.001
Race/Ethnicity	White (non-Hispanic) (%)	68.4	82.9	68.6	<0.001
	Non-Hispanic Black (%)	11.5	8.1	11.2	
	Mexican-American (%)	8.1	2.5	9.0	
	Other Hispanic (%)	5.2	3.0	5.4	
Body mass index (kg/m^2^)		28.5 ± 0.1	29.1 ± 0.1	28.7 ± 0.1	<0.001
Serum Triglycerides (mg/dL)		155.8 ± 3.0	179.3 ± 3.9	152.3 ± 2.3	<0.001
Serum Total cholesterol(mg/dL)		196.6 ± 0.7	192.9 ± 1.0	196.5 ± 0.8	0.096
Serum High density lipoprotein (mg/dL)		53.0 ± 0.5	53.2 ± 0.4	53.1 ± 0.2	0.483
Serum CRP (mg/dL)		0.33 ± 0.03	0.55 ± 0.02	0.29 ± 0.01	<0.001
Fasting blood glucose (mg/dL)		99.3 ± 0.2	113.1 ± 0.8	97.6 ± 0.3	<0.001
Hypertension (%)		15.4	34.7	13.7	<0.001
Diabetes (%)		8.9	21.5	7.8	<0.001
MUFA intake(gm/d)		27.3 ± 0.6	25.6 ± 0.9	27.9 ± 0.6	<0.001
PUFA intake(gm/d)		15.6 ± 0.8	14.9 ± 1.1	16.1 ± 0.4	<0.001

CKD: chronic kidney diseases; CRP: C-reactive protein; Continuous values are expressed as a mean ± SEM.

**Table 2 nutrients-14-01231-t002:** Age, sex, race, fasting blood glucose, systolic and diastolic blood pressure, energy intake, red meat intake, body mass index, diabetes, and hypertension—adjusted mean of markers of kidney function across quartiles of monounsaturated fatty acids and polyunsaturated fatty acids consumption.

**Variables**	**Quartiles of Monounsaturated Fatty Acid (MUFA) Consumption (gm)**					**Quartiles of Polyunsaturated Fatty Acid (PUFA) Consumption (gm)**				
	**1**	**2**	**3**	**4**	***p*-Value ^a^**	**1**	**2**	**3**	**4**	***p*-Value ^a^**
Median (25th–75th)	11.4 (8.4–14.0)	20.9 (18.7–23.0)	30.5 (27.6–33.5)	47.1 (41.5–56.2)		6.2 (4.4–7.7)	11.7 (10.4–13.0)	17.6 (16.0–19.6)	28.8 (24.8–35.9)	
Serum Creatinine (mg/dL)	0.79 ± 0.001	0.81 ± 0.003	0.78 ± 0.001	0.81 ± 0.001	0.186	0.91 ± 0.03	0.83 ± 0.04	0.81 ± 0.05	0.76 ± 0.06	<0.001
Log Urea Albumin (ug/mL)	2.16 ± 0.01	2.20 ± 0.02	2.19 ± 0.01	2.10 ± 0.01	0.415	2.23 ± 0.01	2.17 ± 0.03	2.08 ± 0.01	2.04 ± 0.02	<0.001
Glomerular filtration rate (mL/min/1.73 m²)	84.5 ± 0.36	89.6 ± 0.30	85.4 ± 0.41	86.1 ± 0.37	0.359	86.3 ± 0.5	90.2 ± 0.5	91.4 ± 0.3	96.2 ± 0.5	<0.001
Log Albumin-Creatinine Ratio (mg/dL)	2.10 ± 0.01	2.11 ± 0.01	2.10 ± 0.02	2.09 ± 0.01	0.635	2.17 ± 0.01	2.12 ± 0.03	2.11 ± 0.01	2.04 ± 0.02	<0.001

Values expressed as estimated mean and standard error. ^a^
*p*-values for linear trend across quartiles of MUFA and PUFA consumption. Variables were compared across quartiles of MUFA and PUFA consumption using an analysis of co-variance (ANCOVA) test.

**Table 3 nutrients-14-01231-t003:** Adjusted logistic regression to examine the association between quartiles for mono and polyunsaturated fatty acids and the risk of chronic kidney disease (CKD).

	Likelihood of CKD with Different Models					
Variables	Age, sex, race, poverty to income ratio		Age, sex, race, poverty to income ratio, alcohol intake, energy intake, smoking, physical activity, fasting blood glucose, systolic and diastolic blood pressure, HTN, and DM		Age, sex, race, poverty to income ratio, alcohol intake, energy intake, smoking, physical activity, fasting blood glucose, systolic and diastolic blood pressure, HTN, DM, TG and HDL, and CRP	
	Odds Ratio	Lower Bound- Upper Bound	Odds Ratio	Lower Bound- Upper Bound	Odds Ratio	Lower Bound- Upper Bound
MUFA (Q2)	1.06	(0.62–1.49	0.85	(0.61–1.17)	0.76	(0.55–1.09)
MUFA (Q3)	1.10	(0.58–2.13)	0.96	(0.40–2.13)	0.88	(0.35–2.61)
MUFA (Q4)	0.98	(0.50–1.90)	1.02	(0.29–3.96)	0.96	(0.40–2.13)
PUFA (Q2)	1.01	(0.69–1.43)	1.02	(0.76–1.28)	0.97	(0.78–1.20)
PUFA (Q3)	0.76	(0.69–0.83)	0.81	(0.78–0.86)	0.85	(0.61–1.17)
PUFA (Q4)	0.60	(0.40–0.81)	0.73	(0.65–0.84)	0.79	(0.68–0.88)

The first quartile was always used as a reference. Q2: second quartile; Q3: third quartile; Q4: fourth quartile; CKD: chronic kidney disease; HTN: hypertension; TG: triglyceride; HDL: high density lipo-protein; DM: diabetes; CRP: C-reactive protein; MUFA: monounsaturated fatty acids; PUFA: polyunsaturated fatty acids.

## Data Availability

This investigation uses published or publicly available summary data. No original data were collected for this manuscript.

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
