# Peer review of "Association of Dietary Intakes and Genetically Determined Serum Concentrations of Mono and Poly Unsaturated Fatty Acids on Chronic Kidney Disease: Insights from Dietary Analysis and Mendelian Randomization"

_nutrients, 2022, doi:10.3390/nu14061231_

Round 1

Reviewer 1 Report

This is a cross-sectional study using NHANES data to have explored the association of dietary intakes of mono and poly unsaturated fatty acids with several biomarkers of chronic kidney disease. Given the cross-sectional design and the well-established effect of unsaturated fatty acids on kidney function, this study didn’t add much to the current literature.

On the other hand, this study used a Mendelian Randomization design to have examined how the genetic determined fatty acids level is associated with the kidney function biomarkers. I have a major concern for this part. Because serum fatty acids level is to a large extent determined by dietary intake. So we are not sure how solid it will be to use only genotype data as instrument to predict serum fatty acids level. Particularly given the authors found inconsistent association with kidney functional biomarkers of dietary intakes of fatty acids and the genetic-determined fatty acids level, we may want to question if serum level of nutrients are not quite appropriate for Mendelian Randomization design.

As a minor, the writing of this manuscript can be clearer. Because the current has many description of lab procedures that are not very relevant to the present study.

Also, the abstract can be more fatty acids-kidney disease focused, rather than Mendelian Randomization concentrated. 

Author Response

On the other hand, this study used a Mendelian Randomization design to have examined how the genetic determined fatty acids level is associated with the kidney function biomarkers. I have a major concern for this part. Because serum fatty acids level is to a large extent determined by dietary intake. So we are not sure how solid it will be to use only genotype data as instrument to predict serum fatty acids level. Particularly given the authors found inconsistent association with kidney functional biomarkers of dietary intakes of fatty acids and the genetic-determined fatty acids level, we may want to question if serum level of nutrients are not quite appropriate for Mendelian Randomization design.

Response: Thank you for this very relevant commentary. We have added a statement to draw attention to this possibility in the discussion section of the manuscript.

Changes in manuscript:

“The overall composition of these healthier dietary patterns, as well as other healthy lifestyle behaviours that often occur together [58,61] and which may promote better kidney function, may confound a dietary analysis focused on specific nutrients, in this case PUFAs. Indeed, this may explain the difference in results for the dietary analysis and mendelian randomization in relation to PUFAs and risk of CKD. Similarly, reverse causality, whereby the knowledge of a disease status or disease-marker influences dietary choices, can be particularly problematic in retrospective studies [62]. However, another possible reason for the discrepancy in the dietary analysis and MR results should be considered. It should not be overlooked that serum fatty acids are known to be determined largely due to dietary intake [63]. Therefore, this study may highlight the fact that MR may not be a suitable analysis method for determining the role such serum markers that are more dependent on diet as opposed to genetics.” (Page 9, Lines 333-344 of the revised manuscript).

As a minor, the writing of this manuscript can be clearer. Because the current has many description of lab procedures that are not very relevant to the present study.

Response: Thank you for this important comment. Due to the nature of the analyses performed as well as the combination of dietary intake analysis and MR analysis, along with the multiple models for logistic regression, our statistician feels that the inclusion of this data is important for understanding the context of the results of this study. As such, we respectfully desire to maintain the content of the materials and methods section, please.

Also, the abstract can be more fatty acids-kidney disease focused, rather than Mendelian Randomization concentrated.

Response: Thank you for this valuable suggestion. Due to word-count restrictions with this journal, we are unable to add any further information to the abstract. However, we have added some further information on the role of fatty acids in CKD to the manuscript introduction.

Changes in manuscript:

“Both low plasma concentrations and dietary intakes of n-3 and n-6 polyunsaturated fatty acids (PUFA) have previously been associated with impaired renal function [15-17] while a lower saturated fatty acid (SFA) intake has been associated with improved renal function [18]. The association between monounsaturated fatty acids (MUFA) and risk of CKD, however, remains poorly understood. A potential mechanism by which PUFAs may play a protective role in kidney function by downregulating certain aspects of the inflammatory response, for example, a reduction in proinflammatory cytokines [19]. Indeed, lower levels of plasma markers of chronic inflammation such as CRP and tumor necrosis factor alpha have been observed in older adults with higher serum levels of PUFAs [20]. However, such observational data cannot be used to determine the causality of serum PUFAs in the etiology of CKD.” (Page 2, Lines 49-59 of the revised manuscript).

We are very grateful for these suggestions you have offered and believe they have greatly improved the quality of our manuscript. We hope that you will find our responses and amendments satisfactory and consider our revised manuscript for publication in your journal.

Kindest Regards

Richard Kirwan (on behalf of all the authors)

Reviewer 2 Report

I have two suggestions for Authors, 

Graphical presentation on the flowchart of the study design would improve the clarity the acceptance of the entire study.

Minimum sample size calculation would show the professional approach of Authors. 

Author Response

Graphical presentation on the flowchart of the study design would improve the clarity the acceptance of the entire study.

Response: Thank you for this valuable suggestion. We have prepared a graphical summary of the methodology and results and included it in the updated manuscript

Changes in manuscript:

“Figure 1. Graphical summary of study methodology and results. eGFR: estimated glomerular filtration rate, NHANES: Nutrition and Health Examination Survey, PUFA: polyunsaturated fatty acid. MUFA: monounsaturated fatty acid, CKD: chronic kidney disease, GWAS: genome-wide association study, GPO: genetic predictors of outcomes” (Page 8, Lines 286-289 of the revised manuscript).

Minimum sample size calculation would show the professional approach of Authors.

Response: Thank you for this important comment. For this analysis we have used the largest GWAS that is currently available in the literature and the calculated F-statistics make significant bias from use of weak instruments unlikely. However, we agree that future GWAS with a greater sample size may indeed be warranted and we have amended our limitations section to reflect this.

Changes in manuscript:

“Finally, while we have used the largest GWAS that is currently available in the literature for this analysis, the availability of future GWAS with a greater sample size, and thus providing more statistical power, may warrant further analysis of this topic at that time.” (Page 9, Lines 359-362 of the revised manuscript).

We are very grateful for these suggestions you have offered and believe they have greatly improved the quality of our manuscript. We hope that you will find our responses and amendments satisfactory and consider our revised manuscript for publication in your journal.

Kindest Regards

Richard Kirwan (on behalf of all the authors)